# Association between in-scanner head motion with cerebral white matter microstructure: a multiband diffusion-weighted MRI study

Xiang-zhen Kong

State Key Laboratory of Cognitive Neuroscience and Learning & IDG/McGovern Institute for Brain Research, Beijing Normal University, Beijing, China
Center for Collaboration and Innovation in Brain and Learning Sciences, Beijing Normal University, Beijing, China

Corresponding author
Xiang-zhen Kong,
kongxiangzheng@gmail.com

## ABSTRACT

Diffusion-weighted Magnetic Resonance Imaging (DW-MRI) has emerged as the most popular neuroimaging technique used to depict the biological microstructural properties of human brain white matter. However, like other MRI techniques, traditional DW-MRI data remains subject to head motion artifacts during scanning. For example, previous studies have indicated that, with traditional DW-MRI data, head motion artifacts significantly affect the evaluation of diffusion metrics. Actually, DW-MRI data scanned with higher sampling rate are important for accurately evaluating diffusion metrics because it allows for full-brain coverage through the acquisition of multiple slices simultaneously and more gradient directions. Here, we employed a publicly available multiband DW-MRI dataset to investigate the association between motion and diffusion metrics with the standard pipeline, tract-based spatial statistics (TBSS). The diffusion metrics used in this study included not only the commonly used metrics (i.e., FA and MD) in DW-MRI studies, but also newly proposed inter-voxel metric, local diffusion homogeneity (LDH). We found that the motion effects in FA and MD seems to be mitigated to some extent, but the effect on MD still exists. Furthermore, the effect in LDH is much more pronounced. These results indicate that researchers shall be cautious when conducting data analysis and interpretation. Finally, the motion-diffusion association is discussed.

## INTRODUCTION

Diffusion-weighted MRI (DW-MRI) has become one of the most popular MRI techniques in brain research, as well as in clinical practice. One key application of DW-MRI is diffusion tractography which can be used for the visualization of white matter (WM) tracts (*Golby et al., 2011*) and construction of the brain neuroanatomical connectome (*Gong et al., 2009*). Also, it has become a convenient tool for deriving regional measures of diffusivity and anisotropy. These metrics are believed to reflect biological microstructural properties of the white matter, and have been extensively applied as biological markers for

studying WM under normal and clinical conditions (*Johansen-Berg, 2010*; *Le Bihan, 2003*; *Le Bihan et al., 2001*; *Travers et al., 2012*).

However, like other MRI techniques, DW-MRI remains subject to specific biological factors (e.g., temperature), uncertainty from the scanner (e.g., machine SNR, field shim) and, in particular, motion artifacts. Thus, movement of the head during scanning is undesirable, since it not only displaces the brain matter in space but also interferes with the readout of MR signals. Indeed, recent studies have discovered that head motion may introduce unwanted biases. Ling and his colleagues have shown that head motion is associated with both fractional anisotropy (FA) and mean diffusivity (MD) (the effect is greater for MD) (*Ling et al., 2012*). A recent study have also found group differences in head motion can induce group differences in white matter tract-specific diffusion metrics, and such effects can be more prominent in some specific tracts than others (*Yendiki et al., 2013*). However, these studies on head-motion artifacts have employed traditional DW-MRI data with relatively low sampling rate (e.g., 9 s) and hence few gradient directions (e.g., $n = 30$). In fact, previous works have indicated that more unique sampling directions may decrease bias of diffusion metrics (e.g., FA and MD) (*Landman et al., 2007*; *Tijssen, Jansen & Backes, 2009*). Recently, several promising imaging techniques have been proposed, including MR-encephalography (*Zahneisen et al., 2011*) and multiband echo planar imaging (*Moeller et al., 2010*). Using the multiband scanning protocol, sampling across the whole brain at any given time is allowed through the acquisition of multiple slices simultaneously. Hence, additional gradient directions can be acquired in the same scan duration without loss in spatial resolution. Both of the advantages appear to result in evaluating diffusion metrics more accurately, but little is known about the head motion effects on diffusion metrics from the multiband dataset.

Here, the primary aim was therefore to investigate the relationship between head motion and diffusion metrics estimated from the multiband dataset. In this study, we examined the two tensor-based metrics most typically reported (i.e., FA and MD). Given the fact that they only reflect diffusion properties solely within the voxel, we also examined a newly proposed model-free inter-voxel metric, referred to as local diffusion homogeneity (LDH) (*Gong, 2013*). We hypothesized that motion effects would be mitigated in the multiband DW-MRI data. It has been suggested that head motion alters the measure of diffusion metrics even after motion and eddy current correction (*Ling et al., 2012*; *Tijssen, Jansen & Backes, 2009*; *Yendiki et al., 2013*), and that it may also provide information regarding neuronal processing (*Yan et al., 2013a*; *Yan et al., 2013b*). Moreover, LDH is a recently proposed metric and has not been fully validated yet (*Gong, 2013*). In addition, unlike FA and MD, LDH directly depends on the raw diffusivity series without assuming a prior diffusion model (*Gong, 2013*). Therefore, we also hypothesized that the association between head motion and LDH would be quite different to the tensor-based metrics, and may be more sensitive to motion artifacts. We tested these hypotheses by (1) confirming the test-retest reliability of both diffusion metrics and head motion across scan sessions, and (2) by examining the relationship between the averaged diffusion metrics and head motion. We also examined the relationship in each scan session.

## MATERIALS AND METHODS

### Dataset

The dataset used in this study was from the NKI-RS Multiband Imaging Test-Retest Pilot Dataset (*Mennes et al., 2013*). There were 20 participants (34.3 ± 14.0 years). For each participant, the DW-MRI scans were performed twice (session 1 and session 2), around one week apart. Diffusion weighted images were collected a standard pulse sequence with 2-mm-thick axial slices and 137 directions: TE 85 ms; TR 2400 ms; b value, 1500 s/mm$^2$; flip angle, 90°.

### Image processing

DW-MRI images were processed with FMRIB's Software Library (FSL, http://www.fmrib.ox.ac.uk/fsl). Non-brain tissue was removed using the Brain Extraction Tool (BET) with a fractional intensity threshold of 0.2, and then raw DW images were affinely registered to the nonDW image, to partially correct for the effects of motion and eddy currents. Then, by fitting a tensor model at each voxel using DTIFit from the FSL (*Smith et al., 2004*), we obtained the fractional anisotropy (FA) and mean (MD) diffusivity, used in subsequent TBSS analysis (*Smith et al., 2006*; *Smith et al., 2007*).

To compare between subjects, the TBSS framework was used. In detail, first, we non-linearly aligned the individual FA maps to FSL's standard 1 mm isotropic FA template (FMRIB58_FA) and averaged them to generate a study specific mean FA map. Next, voxels with an FA > 0.2 in the mean FA map were masked out, and the reminder was thinned to create a white matter "skeleton". The resulting skeleton contained WM tracts common to all subjects. Individual FA maps were then projected onto the mean FA skeleton by filling the skeleton with FA values from the nearest tract center. The same non-linear transformations derived for the FA maps were applied to the MD maps.

In terms of the LDH metric, it is a novel model-free metric that defines the regional inter-voxel coherence of diffusion series (*Gong, 2013*). Technologically, LDH is quantified within the neighbors ($n = 27$) via the Kendell's coefficient concordance (KCC), after the estimation of the diffusivity strengths along each gradient direction. To compare between subjects, the LDH maps were also projected onto the WM skeleton mask using the TBSS framework described above. In addition, we used the same approach with different neighbor size (i.e., $n = 7$ and $n = 19$) for quantifying the LDH, and also used another approach for quantifying the regional coherence with information theory (*Kong, Zhen & Liu, 2014*). The results in these cases were all similar to those of the original LDH (data not shown).

The DW-MRI data preprocessing and TBSS analysis pipelines were both implemented using Nipype (*Gorgolewski et al., 2011*), a flexible, lightweight and extensible neuroimaging data processing framework in python. The pipeline for calculating both original and improved LDH was implemented in Python.

## Assessment of in-scanner head motion

To retrospectively estimate head motion during scanning, DW images were realigned to the non-DW image with FMRIB's Linear Image Registration Tool (FLIRT), and at the same time, an affine transformation matrix was obtained for each image. Then, for each image, the root mean squared (RMS) deviation (*Jenkinson et al., 2002*), a summary statistic of in-scanner head motion, was calculated from its transformation using the tool *rmsdiff* from FSL. Since it summarizes six translational and rotational parameters, the RMS has been widely used in the neuroimaging community. For instance, it has been used in fMRI and DTI data processing to check the extent of head motion and make decisions about cohort formation or matching (e.g., *Ikuta et al., 2014*; *Kochunov et al., 2013*; *Kong, 2014*). Technically, the RMS can be calculated directly from the affine matrices with the formula (1).

$$\text{RMS} = \sqrt{\frac{1}{5}R^2\text{Trace}(A^T A) + t^T t} \tag{1}$$

In formula (1), $R$ is a radius specifying the volume of interest ($R = 80$ mm, approximately the mean distance from the cerebral cortex to the center of the head), A is a $3 \times 3$ 'rotation' matrix and t is a $3 \times 1$ column vector of translation. One thing to note is that since the RMS uses all the information from the affine matrices (including the shear and scaling, if present), it could include the electrical properties of each participant's head. Nevertheless, the RMS does provide a sensitive index of in-scanner head motion.

Here, the RMS was calculated from 2 transformations of consecutive images (*Jenkinson et al., 2002*). That is, in-scanner head motion was measured as the summary measure of head motion relative to the preceding volume as the previous studies (*Satterthwaite et al., 2012*; *Van Dijk, Sabuncu & Buckner, 2012*). Finally, head motion was calculated by averaging the RMS deviations for all volumes.

## Test-retest reliability of diffusion metrics and head motion estimate

The voxel-wise test-retest reliability for each diffusion metric was calculated with the intra-class correlation coefficient (ICC) (*Shrout & Fleiss, 1979*).

$$\text{ICC} = \frac{\text{BMS} - \text{EMS}}{\text{BMS} + (k-1)\text{EMS}} \tag{2}$$

The formula estimates the correlation of the subject signal intensities between sessions, modeled by a two-way ANOVA, with random subject effects and fixed session effects. In this model, the total sum of squares is split into subject (BMS), session (JMS) and error (EMS) sums of squares; the $k$ is the number of repeated sessions. The reliability measure for whole-brain analysis was implemented in python and can be accessed from Nipype (*Gorgolewski et al., 2011*). The test-retest reliability for head motion estimate was also calculated with ICC.

**Table 1 A basic summary of head motion and the motion effects in three diffusion metrics.** The column Motion includes the averaged motion in the sample. The column Motion-Brain Association includes the summary of motion effects in different diffusion metrics (i.e., FA, MD, and LDH). n.r. indicates null results; Plus sign (+) indicates a positive relationship, while minus (−) indicates a negative relationship.

| Sample | Motion | Motion-Brain Association | | |
|---|---|---|---|---|
| | | FA | MD | LDH |
| Session 1 | 1.1(0.32) | n.r. | +[*] | −[*] |
| Session 2 | 1.26(0.38) | n.r. | +[*] | −[**] |
| Averaged | 1.18(0.29) | n.r. | +[**] | −[**] |

**Notes.**

[*] $p < 0.10$ TFCE corrected.
[**] $p < 0.05$ TFCE corrected.

### Relationship between in-scanner head motion and diffusion metrics

To maximize the signal to noise ratio of head motion estimates, first we calculated the average head motion for each participant across two sessions. Analogously, for accurate measures of microstructure estimates, the MD, FA and LDH metrics finally used were also taken from the average of the TBSS results across the two sessions. To examine the possible relationship between head motion and diffusion regional metrics, we conducted a statistical analysis using general linear models (GLMs), for the three metrics respectively, with head motion as the variable of interest. In these models, gender, age and handedness were controlled as confounding covariates. Handedness was included here because it is associated with brain structure and the neural processing of attention, while attention deficit may cause more head motion (e.g., *Durston et al., 2003*; *Schmidt, Simões & Schmidt, 2013*). Voxel-wise statistical analysis was performed with Threshold-Free Cluster Enhancement (TFCE) correction (*Smith & Nichols, 2009*) for multiple comparisons, considering fully corrected $p$-value $<0.05$ as significant. In addition, the same statistical procedure was conducted for both session 1 and session 2 respectively.

## RESULTS

### Test-retest reliability of diffusion metrics and head motion estimate

All of the diffusion metrics in this study showed relatively high test-retest reliability: FA: Mean ICC $= 0.71$; MD: Mean ICC $= 0.71$; LDH: Mean ICC $= 0.75$. In addition, the magnitude of head motion seemed acceptable (Table 1) and showed medium reliability (ICC $= 0.54$), which is consistent with previous studies (*Van Dijk, Sabuncu & Buckner, 2012*). Although all of the diffusion metrics, as well as the head motion estimate, are relatively reliable across the two scans, they were not exactly the same due to some random artifacts, including motion artifacts and machine noises. Thus, for accurate measures of this microstructure and the head motion estimate, we first averaged the head motion

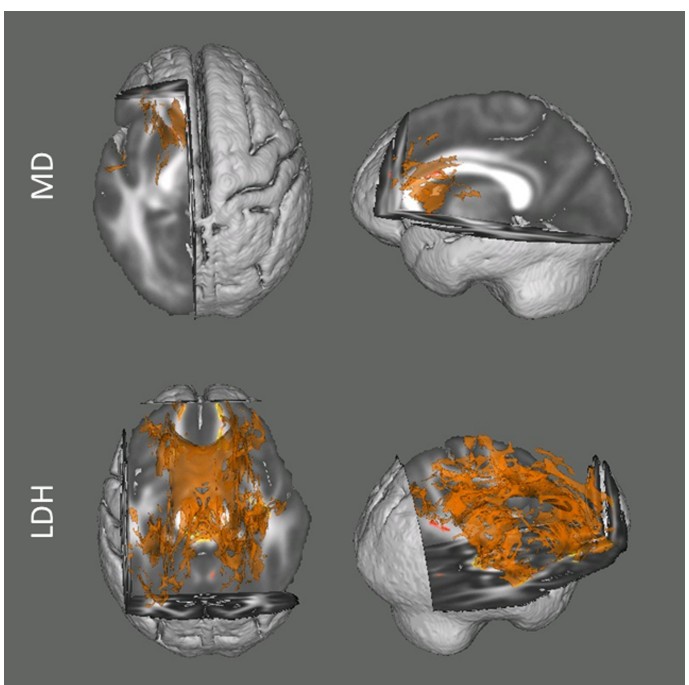

**Figure 1 Results from the tract-based spatial statistics (TBSS) analyses depicting the voxels that showed significant associations between head motion and diffusion metrics.** Data are presented for the analyses involving both Mean Diffusivity (MD; A) and Local Diffusion Homogeneity (LDH; B) as the dependent measure. Participants with higher motion exhibited higher apparent values of MD, but lower LDH. Voxels survived the TFCE correction ($p < 0.05$) across the whole white matter skeleton are displayed.

and diffusion metrics across the two sessions and mainly examined the results with the averaged data.

## Relationship between the head motion estimate and diffusion metrics

In Table 1, we also show a basic summary of the main head motion results of diffusion metrics across all three analyses.

Among the two mostly commonly used regional diffusion metrics (i.e., FA and MD), these results indicated that head motion was mainly associated with the MD values. The degree of head motion was positively associated with increased MD mainly within white matter tracts in left hemisphere, including anterior limb of internal capsule, posterior limb of internal capsule, genu of corpus callosum and body of corpus callosum (Fig. 1). In the current report, we focus on voxels that survived the TFCE correction ($p < 0.05$) (*Smith & Nichols, 2009*) within the whole white matter skeleton. For the analyses examining FA, no voxel survived correction for multiple comparisons.

For the analyses examining the inter-voxel diffusion metric (i.e., LDH), we found that wide-spread white matter showed significant negative association with head motion ($p < 0.05$, TFCE corrected; Fig. 1). This association mainly involved the bilateral superior longitudinal fasciculus, body and genu of corpus callosum, cingulum, superior, anterior

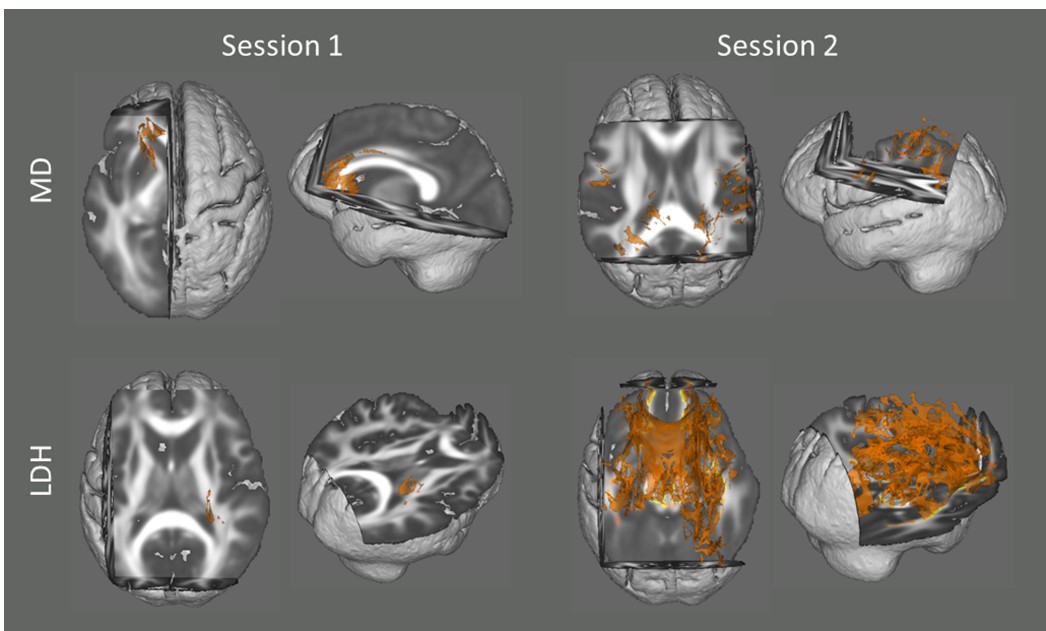

**Figure 2 Results from the tract-based spatial statistics (TBSS) analyses depicting the voxels that showed significant associations between head motion and diffusion metrics.** Since no voxel survived statistical correction for multiple comparisons ($p < 0.05$) in most of the analyses (except LDH in Session 2), they are displayed at a more tolerant threshold ($p < 0.10$, TFCE corrected).

and posterior corona radiate, retrolenticular part of internal capsule, fornix, cerebral peduncle, middle cerebellar peduncle, right anterior and posterior limb of internal capsule, right external capsule and sagittal stratum.

Overall, with the same criterion of significance ($p < 0.05$, TFCE corrected), we found the most number of motion-related voxels with LDH (34551 voxels, 32.25%) and then MD (2686 voxels, 2.51%). No voxel survived with FA.

In addition, we also examined the association between head motion and diffusion metrics with data from the two sessions respectively. Although no voxel survived statistical correction for multiple comparisons ($p < 0.05$) in most of the analyses (except LDH in Session 2), there were some voxels that showed a significant trend ($p < 0.10$, TFCE corrected; Fig. 2).

## DISCUSSION

Like any other MRI technique, DW-MRI signal is subject to head motion artifacts, however, the relationship between diffusion metrics and head motion remains incompletely understood. Previous studies have shown a significant relationship between diffusion metrics and head motion (*Ling et al., 2012*) with conventional scanning protocol. The current study expands on previous work by exploring the relationship between motion and diffusion metrics (including the recently proposed inter-voxel metric, LDH) with a multiband dataset. We found that the motion effects in FA and MD seems to be mitigated to some extent, but the effect on MD still existed. In addition, the effect is much more pronounced in LDH. Since these results are present following standard
processing procedures, researchers shall be cautious when conducting data analysis and interpretation.

Previous studies suggested a positive association between motion and MD, with increased magnitude of MD as a result of increased total motion (*Ling et al., 2012*). The results of this study, with multiband dataset, replicate this finding, as a positive relationship between head motion and the magnitude of MD was present in the left hemisphere tracts. The significant association was mainly located in the deeper white matter (e.g., corpus callosum and the internal capsule). Interestingly, these tracts have often been reported in the literature to differ between a variety of clinical populations and healthy subjects (*Carrasco et al., 2012*; *Travers et al., 2012*). For examining FA, we found no significant relationship between head motion and FA after multiple comparison correction. On the one hand, our findings appear consistent with the previous finding (*Ling et al., 2012*) that the head motion's bias is more pronounced in MD than FA. But on the other hand, given the reduction of the number of motion-related voxels (multiband dataset: 0 voxel for FA, 2686 voxels for MD; *Ling et al. (2012)*: 2422 voxels for FA, 22679 voxels for MD), the motion effect seems to be mitigated in the multiband dataset. This may be due to several advantages of the multiband dataset. First, the multiband dataset was acquired with much more gradient directions than traditional datasets, which would result in more accuracy when evaluating diffusion metrics. Second, the multiband scanning protocol allows full-brain coverage through the acquisition of multiple slices simultaneously. This could avoid displacements of brain within a TR and further mitigate the motion effects. Finally, given the fact that multiband protocol is designed for a relatively high sampling rate (i.e., a shorter TR), motion effects from a shorter duration would be expected to decrease. All these advantages could result in higher accuracy and less irrelevant effects (e.g., head motion) when evaluating diffusion metrics.

In addition, we also explored the relationship between head motion and LDH values and found that there were widespread voxels significantly associated to head motion. It's worth noting that with a smaller neighbor sizes ($n = 19$ or $n = 7$) when calculating the LDH, we observed the similar result (number of voxels that survived the TFCE correction: 35570 voxels for $n = 19$; 40 425 voxels for $n = 7$). This appears to be quite likely caused by motion artifacts. Indeed, the significant voxels were rarely located in the occipital white matter tracts, where motion artifacts may be much weaker than that in prefrontal lobe when subjects are laying supine. These results suggest that LDH values might be more subject to head motion artifacts. The increased susceptibility to motion may be due to the fact that it is an inter-voxel metric which would be subject to shear in the displacement, and that it is directly calculated with raw diffusivity series. Though previous studies have shown LDH values change during aging, the newly proposed metric has not yet well validated (*Gong, 2013*) and simulation and experimental work is required to confirm the motion-LDH association.

So, why does the association between motion and diffusion metrics exist? The dominant view at present is that head motion introduces artifacts into diffusion signals, similar to

what has been noted in the fMRI literature (*Bullmore et al., 1999*; *Friston et al., 1996*; *Hajnal et al., 1994*), which influence the calculation of diffusion metrics and further results of cross-subject analysis. A common strategy for controlling motion effects in neuroimaging cross-subject analysis is to regress or match motion estimates (*Zuo et al., 2010a*; *Zuo et al., 2012*; *Zuo et al., 2010b*). Another strategy for mitigating head-motion artifacts is to remove time series of high motion, which is called 'scrubbing' (*Power et al., 2012*). However, these strategies have their limitations. On the one hand, scrubbing volumes with high motion could not fundamentally change the relationship between motion and values of diffusion metrics (*Ling et al., 2012*). On the other hand, they may also reduce the ability to detect a significant effect of interest, and/or introduce sampling bias (*Satterthwaite et al., 2012*; *Wylie et al., 2014*).

While researchers attempt to propose more sophisticated algorithms, there is growing perception in the field that head motion reflects individual differences in psychological traits and clinical conditions. For instance, previous studies showed that head motion was correlated with some psychological and clinical measures, such as the autism symptom severity score (*Yendiki et al., 2013*). In addition, previous fMRI studies suggest that the association may reflect the neural processing related to head motion (*Yan et al., 2013a*; *Yan et al., 2013b*). However, it is important to note that this problem in dMRI would not be as serious as it is in fMRI, since in dMRI the neural processing causing the motion does not directly affect the signal intensity. Nevertheless, these findings do suggest that head motion might not simply be an uncorrelated random variable.

Taken together, as articulated previously (*Van Dijk, Sabuncu & Buckner, 2012*; *Wylie et al., 2014*; *Yendiki et al., 2013*), these findings demonstrate the significance of developing motion-compensated acquisition methods for DW-MRI and incorporating them into neuroimaging studies in the future. Nevertheless, with current technologies, it appears impossible to perfectly eliminate the motion effects. As a temporary solution, examining both models with and without motion being regressed out will be expected. But in this case, researchers should include both results in the report, rather than just pick a 'better' one. Additionally, researchers shall keep in mind that motion does not only influence MRI signals, but also correlated with some meaningful individual differences. Alternatively, replication in an independent sample would be helpful, since the effects of head motion on diffusion metrics are usually random and not specific to some brain regions. Nevertheless, for now, researchers shall be cautious when doing MRI data analysis and interpretation.

In sum, the results of this study indicate that, in the multiband diffusion data, there are also significant associations between head motion and diffusion metrics, although the motion effects appear to be mitigated compared to those with traditional datasets. Specifically, head motion was associated with both MD and LDH, and no significant effect was found for FA. Future studies should investigate the association between head motion and diffusion metrics with larger multiband datasets.

### Funding

There was no funding source for this study.

### Competing Interests

The authors declare there are no competing interests.

### Author Contributions

- Xiang-zhen Kong conceived and designed the experiments, performed the experiments, analyzed the data, contributed reagents/materials/analysis tools, wrote the paper, prepared figures and/or tables, reviewed drafts of the paper.

### Data Deposition

The following information was supplied regarding the deposition of related data:
Multiband Imaging Test-Retest Pilot Dataset: http://fcon_1000.projects.nitrc.org/indi/pro/eNKI_RS_TRT/FrontPage.html

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
