# Peer review of "Association between in-scanner head motion with cerebral white matter microstructure: a multiband diffusion-weighted MRI study"

_PeerJ, doi:10.7717/peerj.366_

## Round 0.1 · original submission · Major Revisions

Please thoroughly check and correct grammatical errors in the manuscript and address each of the reviewers comments and concerns.

Reviewer 1 ·

Basic reporting

The paper is considering the impact of motion on diffusion related measures. Citing previous literature this has been established and is here reproduced using a multiband sequence, which allows for a 3-fold increase in the number of diffusion volumes. No differentiation of the motion artifacts are described, and they are treated as combination of sources.

The article fails in make it clear that the difference between previous studies and the current is that there is an increase in volumes and that all other aspects are the same. This oversight has the implication that the conclusions wrt. increased SNR are not clear, but are instead attributed to a sequence.

Experimental design

The question is the impact of SNR on evaluating the motion vs. diffusion measure issue. This is masked through making the statements appear as being an evaluation of a sequence. Indeed the volume acquisition with the multiband is about 1/3 of the conventional, but the sensitivity to motion during the diffusion encoding is still the same for both sequences.

Validity of the findings

The findings are acceptable.

Additional comments

abstract: "subject to motion", clarify which motion sources are being considered
"without significant loss in spatial resolution". The multiband does not have a loss in resolution but some loss in SNR.
Page 4 line 37: "movement of the head during scanning". Is this motion between volumes, during the diffusion weighting
Page 4, line 49. No loss in spatial resolution
page 4, line 49. If the source is from limited number of direction, it should be reduced with 3X the directions. If it is motion, then it should stay the same
Page 5, line 57. The diffusion model has to corrected to take account of the changes in local diffusion due to motion (variation due to gradient non-linearity etc.).
page 5, line 76. This step can introduce a lot of bias
Page 9, line 175-178. This has implications for what the study can tell. If SNR is the issue for FA, what is it for MD
Line 199. A reference to a paper under review is not acceptable
Line 205. The un-interesting motion should be compensated, but the important motion should stay

Reviewer 2 ·

Basic reporting

The writing has many grammatical errors, and in places appears to leave out some important information. Because of the resulting ambiguity, I was not able to judge the validity of the paper's content as well as I would like. I encourage the author to find a local researcher proficient in English who could help ensure that all of the concepts are clearly presented.

L12: Diffusion MRI has been around for two decades, so “promising” is no longer necessary.
L13: Remove “the”.
L16: relative -> relatively
L17: improves -> improve
L19: remove “a”
L22: “The” -> “A”, and “while the” -> “and a”
L24 (or the abstract in general): The statement is fine in itself, but the abstract is vague about why the study was done. Was motion expected to have less effect on multiband data because there was less time per volume, and/or more volumes?
L30: brain -> the brain
L36: scanner noises -> uncertainty from the scanner
L37: which -> since it
L40: while the effect is greater for MD -> (the effect is greater for MD)
L41: have -> has
L43: on -> of
L44: small number -> few
L45: The “influence” being referred to needs to be specified in the text.
L52: two -> the two
L55: “be more or less related” -> “affect the” ?
L62: tenser -> tensor
L69: “And for” -> “For”
L75: factional -> fractional
L76: It would be more accurate to say “then the DW images were affinely registered to the nonDW image, to partially correct for the effects of motion and eddy currents”.
L80: “TBSS framework” -> “the TBSS framework”, aligned -> aligned the
L81: map -> maps
L82, 83: The step order is reversed. It should be “Next, voxels with an FA < 0.2 in the mean image were masked out, and the remainder was thinned to create...”
L106: metrics -> metric
L115: signal -> the signal
L119: “general linear model (GLM),” -> “general linear models (GLMs)”. (Unless there really was just one GLM. If so, please explain.)
L120: “handiness that were available in the dataset” -> “handedness” (As available? If these variables were not uniformly available, explain. I am assuming you mean handedness since that is more likely to be available than handiness.)
L127: relative -> relatively. Something seems wrong with the reported means. Are these mean ICCs? (As opposed to mean FAs, etc.)
L130: “head motion estimate” -> “head motion estimates” or “the head motion estimate” (?), “is relative” -> “are relatively”
L131: exact -> exactly
L132: “this microstructure and head motion estimate” -> “the microstructure and head motion estimates” (?)
L135: “head motion estimate” -> “the head motion estimates” (?)
L142: Insert “which” or “that” before survived.
L143: I am confused about the connection between individual voxels and “across the whole white matter skeleton”. Clarify.
L146: Insert “which” or “that” before showed?
L147: bilateral -> the bilateral
L153: Switch the order of LDH and MD.
L157: Insert “which” or “that” before showed.
L163: significant -> a significant
L165: newly -> recently
L166: association -> associations
L170: I do not understand why you mention “one axes”, then all three.
L171: “Current results” -> “The results of this study”?
L172: magnitude -> the magnitude. (or just remove “magnitude of”). mainly -> is mainly
L177: “But this” -> “This”
L181: rarely -> were rarely
L182: laying -> are
L183-184. Do not start a sentence with “And”, and change “susceptibility” to “susceptibility to motion” since susceptibility by itself has a special meaning in magnetism.
L188: similar to be -> similar to what has been
L207: remove “not”
L208: The recommendation introduces a multiple comparisons problem. Do not start a sentence with “But”.
L212: “shall” -> “should”. I understand what you are saying, but I think it would be worth emphasizing that the problem in dMRI is not as serious as it is in fMRI. In dMRI the structures affected by motion in the images are not necessarily the ones causing the motion. In fMRI the voxel intensity maps are also affected by activation when the subject moves.
L213: “current results” -> “the results of this study”. Again, you need to explain what it is you expect the results to be relevant for. Faster acquisition per volume? More directions? Something more specific to multiband excitation?
L215: remove “be to”
Table 1: The caption needs to be rephrased. The table itself is also confusing, since there are multiple levels, and presumably kinds, of averaging being done, e.g. Mean MRD expands to Mean Mean Relative Displacement. It would be more customary to put the Averaged sample at the bottom of the table.

Figure 1: Are the above-threshold voxels shown in a single color (i.e. a binary overlay), or with a continuous colormap indicating significance level? Either way is fine, I am simply curious.

Figure 2: “diffusion” -> “and diffusion”, and insert “which” or “that” before survived. “higher values of MD, but lower values of LDH” -> “higher apparent values of MD, but lower LDH.”

Experimental design

The data was already published, so I will focus on the analysis methods. Since the article needs extensive revision, my comments here are not complete.

L98-104: Please provide more detail.
For the purpose of motion estimation (as opposed to alignment for tensor estimation), were the DW images aligned to the nonDW image with affine or rigid registration? If it was affine, what method was used to extract the rigid (esp. rotation) part of the affine matrix? If it was rigid, then what about the effect of eddy currents?
The deviation (rightly so) should include the effect of rotation, but the most natural interpretation of “the displacement of each brain volume relative to the preceding one” is a simple translation. It would make more sense to me if “volume” were replaced with “voxel” throughout the last two sentences of this paragraph. Would “The head motion of a volume was calculated as the RMS displacement between each voxel and the corresponding location in the previous volume.” accurately describe what was done? If so, were extraneous voxels (e.g. air, skull(?)) masked out? Including air would introduce edge effects, and features outside the braincase are not necessarily rigidly linked to the brain.

Validity of the findings

My only concern at this point is that I was not able to completely follow each step of the analysis. I do not really doubt the findings, but reserve final judgement for a future revision.

L183-184: An even more likely reason why LDH is affected by motion is that it is an intervoxel measure, so it is subject to shear in the displacement field, and is likely more prone to resampling effects. Gong’s paper allows the number of neighbors to be varied – there might be less sensitivity to motion if the number of neighbors is reduced.

Additional comments

Thank you for doing this study. Although it needs extensive rewriting, my impression is that most of the work to be done is in the writing, and the analysis will need little additional work before publication. Technically the motivation for the study was given, but it seems that multiband dMRI was expected to be less(?) affected by motion than traditional DTI. Instead of forcing the reviewers and readers to speculate on the motivation(s), the initial expectations should be given in the abstract and discussed in the Discussion section.

---

## Round 0.2 · Minor Revisions

Please address these few remaining comments, particularly the reviewers issue concerning the experimental design.

Reviewer 2 ·

Basic reporting

Line 44: remove "number of".

Line 48: This is confusing because single band acquisitions also allow the eventual coverage of the full brain. Replace "full-brain coverage" with "sampling across the whole brain at any given time".

Line 105: The formula should be explicitly given here as in lines 6 to 7 of Kong2014a page 3, with the addition of the caveat about shear being included in the affine transformations.
If you did use the formula in Kong2014a for this study, then the rest of my 12th comment (brain extraction, etc.) are largely irrelevant because the formula approximates the brain with a sphere.

Since there will then be more than one equation, they should be numbered.

Line 221: Replace "structures ... processing" with "neural processing causing the motion does not directly affect the signal intensity".

Line 222: Replace "just a" with "simply be an uncorrelated".

Experimental design

Line 102 (+ author's reply to my 12th comment in the original review): Affine is not the same as rigid. The affine matrix includes shear due to eddy currents, so a motion measure that treats affine transformations as if they are rigid mixes the effects of eddy currents in with motion. Ideally the affine matrix should be decomposed into rotation, shear, magnification, and translation parts, and then the motion summary should use only the rotation and translation. (The magnification matrix should be close to the identity matrix. If not, that is another problem.) The method used to decompose the affine matrices should be described. If the author is unwilling to redo the calculations, this paragraph will need to be rewritten to clarify that the motion summary does *not* use rigid transformations, and thus includes the electrical properties of each subject's head as a potentially confounding variable.

Validity of the findings

Lines 126 and 127: Do not use a preprint as support of a claim.

Additional comments

I have no further requests, but I am interested in how some of Reviewer #1's points will be handled.

---

## Round 0.3 · accepted · Accept

I have no further comments